# Tolerability and Preliminary Outcomes of Adjuvant T-DM1 in HER2-Positive Breast Cancer After Neoadjuvant Therapy: The ATD Study

**DOI:** 10.3390/cancers16234104

**Published:** 2024-12-07

**Authors:** Eriseld Krasniqi, Lorena Filomeno, Teresa Arcuri, Francesca Sofia Di Lisa, Antonio Astone, Claudia Cutigni, Jennifer Foglietta, Martina Nunzi, Rosalinda Rossi, Mauro Minelli, Icro Meattini, Luca Visani, Jacopo Scialino, Lorenzo Livi, Luca Moscetti, Paolo Marchetti, Andrea Botticelli, Ida Paris, Francesco Pavese, Tatiana D’Angelo, Valentina Sini, Simonetta Stani, Maria Rosaria Valerio, Antonino Grassadonia, Nicola Tinari, Marco Mazzotta, Matteo Vergati, Giuliana D’Auria, Teresa Gamucci, Loretta D’Onofrio, Simona Gasparro, Arianna Roselli, Alberto Fulvi, Gianluigi Ferretti, Andrea Torchia, Monica Giordano, Filippo Greco, Francesco Pantano, Giuseppe Tonini, Agnese Fabbri, Emilio Bria, Giovanna Garufi, Elena Fiorio, Mimma Raffaele, Mirco Pistelli, Rossana Berardi, Rosa Saltarelli, Ramy Kayal, Francesca Romana Ferranti, Katia Cannita, Azzurra Irelli, Nicola D’Ostilio, Costanza De Rossi, Raffaella Palumbo, Anna Cariello, Giuseppe Sanguineti, Fabio Calabrò, Laura Pizzuti, Maddalena Barba, Claudio Botti, Fabio Pelle, Sonia Cappelli, Flavia Cavicchi, Ilaria Puccica, Amedeo Villanucci, Isabella Sperduti, Gennaro Ciliberto, Patrizia Vici

**Affiliations:** 1Phase IV Clinical Studies Unit, IRCCS Regina Elena National Cancer Institute, 00144 Rome, Italy; eriseld.krasniqi@ifo.it (E.K.); lorena.filomeno@ifo.it (L.F.); patrizia.vici@ifo.it (P.V.); 2Medical Oncology Unit, San Giovanni Evangelista Hospital, ASL RM5, 00019 Tivoli, Italy; francescasofia.dilisa@aslroma5.it (F.S.D.L.); rosa.saltarelli@aslroma5.it (R.S.); 3Oncology Division, San Pietro Fatebenefratelli Hospital, 00189 Rome, Italy; antonio.astone@gmail.com (A.A.); claudiacutigni1992@gmail.com (C.C.); 4Oncology Unit, Ospedale Santa Maria, 05100 Terni, Italy; j.foglietta@aospterni.it (J.F.); martina.nunzi@aospterni.it (M.N.); 5Division of Oncology, San Giovanni Addolorata Hospital, 00184 Rome, Italy; rrossi@hsangiovanni.roma.it (R.R.); mminelli@hsangiovanni.roma.it (M.M.); 6Department of Experimental and Clinical Biomedical Sciences “M. Serio”, University of Florence, 50134 Florence, Italy; icro.meattini@unifi.it (I.M.); lorenzo.livi@unifi.it (L.L.); 7Radiation Oncology Unit, Oncology Department, Azienda Ospedaliero-Universitaria Careggi, 50134 Florence, Italy; luca.visani@unifi.it (L.V.); jacopo.scialino@unifi.it (J.S.); 8Oncology and Hemathology Department, Azienda Ospedaliero-Universitaria Policlinico di Modena, 41125 Modena, Italy; moscetti.luca@aou.mo.it; 9Istituto Dermatopatico dell’Immacolata IRCCS, 00167 Rome, Italy; paolo.marchetti@hotmail.it; 10Department of Radiological, Oncological and Pathological Science, Policlinico Umberto I, “Sapienza” University of Rome, 00161 Rome, Italy; andrea.botticelli@uniroma1.it; 11Division of Gynecologic Oncology, Department of Woman and Child Health and Public Health, Fondazione Policlinico Universitario Agostino Gemelli IRCCS, 00136 Rome, Italy; ida.paris@policlinicogemelli.it (I.P.); francesco.pavese@guest.policlinicogemelli.it (F.P.); tatiana.dangelo@guest.policlinicogemelli.it (T.D.); 12Medical Oncology, Santo Spirito Hospital, 00193 Rome, Italy; valentina.sini@aslroma1.it (V.S.); simonetta.stani@aslroma1.it (S.S.); 13Medical Oncology, AOU Policlinico Paolo Giaccone, 90127 Palermo, Italy; valerionc17@gmail.com; 14Department of Innovative Technologies in Medicine and Dentistry, Centre for Advanced Studies and Technology (CAST), G. D’Annunzio University Chieti-Pescara, 66100 Chieti, Italy; grassa@unich.it; 15Department of Medical, Oral and Biotechnological Sciences, Center for Advanced Studies and Technology (CAST), G. D’Annunzio University Chieti-Pescara, 66100 Chieti, Italy; ntinari@unich.it; 16Medical Oncology Unit, Department of Medical Oncology, Sandro Pertini Hospital, 00157 Rome, Italy; marco.mazzotta@aslroma2.it (M.M.); matteo.vergati@aslroma2.it (M.V.); giuliana.dauria@aslroma2.it (G.D.); teresa.gamucci@gmail.com (T.G.); 17Division of Medical Oncology 1, IRCCS Regina Elena National Cancer Institute, 00144 Rome, Italy; loretta.donofrio@ifo.it (L.D.); simona.gasparro@ifo.it (S.G.); arianna.roselli@ifo.it (A.R.); alberto.fulvi@ifo.it (A.F.); gianluigi.ferretti@ifo.it (G.F.); andrea.torchia@ifo.it (A.T.); fabio.calabro@ifo.it (F.C.); laura.pizzuti@ifo.it (L.P.); maddalena.barba@ifo.it (M.B.); 18Clinical and Molecular Medicine, Sapienza-Università di Roma, 00189 Rome, Italy; 19Medical Oncology Department, Azienda Socio Sanitaria Lariana, 22100 Como, Italy; monica.giordano@asst-lariana.it; 20Medical Oncology Unit, Mater Salutis Hospital, Ulss 9 Veneto Region, 37045 Legnago, Italy; filippo.greco@aulss9.veneto.it; 21Department of Medical Oncology, Fondazione Policlinico Universitario Campus Bio-Medico, 00128 Rome, Italy; f.pantano@policlinicocampus.it (F.P.); g.tonini@policlinicocampus.it (G.T.); 22Medical Oncology Unit, Belcolle Hospital, 01100 Viterbo, Italy; agnese.fabbri@yahoo.it; 23Università Cattolica del Sacro Cuore, Fondazione Policlinico Universitario Agostino Gemelli, IRCCS, 00136 Rome, Italy; emilio.bria@policlinicogemelli.it (E.B.); giovanna.garufi@guest.policlinicogemelli.it (G.G.); 24Ospedale Isola Tiberina–Gemelli Isola, 00186 Rome, Italy; 25Section of Oncology, Department of Medicine, University of Verona School of Medicine and Verona University Hospital Trust, 37126 Verona, Italy; elena.fiorio@aovr.veneto.it; 26Presidio Cassia Sant’andrea, Dipartimento Oncologico, Asl Roma 1, 00189 Rome, Italy; mimma.raffaele@aslroma1.it; 27Department of Medical Oncology, Università Politecnica delle Marche, AOU delle Marche, 60126 Ancona, Italy; pistelli.mirco@gmail.com (M.P.); r.berardi@univpm.it (R.B.); 28Radiology Unit, IRCCS Regina Elena National Cancer Institute, 00144 Rome, Italy; ramy.kayal@ifo.it (R.K.); francescaromana.ferranti@ifo.it (F.R.F.); 29Medical Oncology Unit, Department of Oncology, “Giuseppe Mazzini” Hospital, AUSL Teramo, 64100 Teramo, Italy; katia.cannita@aslteramo.it (K.C.); azzurra.irelli@aslteramo.it (A.I.); 30Medical Oncology, ASL 2 Abruzzo, Ospedale Floraspe Renzetti, 66034 Lanciano, Italy; nicola.dostilio@asl2abruzzo.it; 31Medical Oncology Unit, AULSS 3 Serenissima, 30174 Mestre-Venice, Italy; costanza.derossi@aulss3.veneto.it; 32Medical Oncology Unit, Istituti Clinici Scientifici Maugeri IRCCS, 27100 Pavia, Italy; raffaella.palumbo@icsmaugeri.it; 33Oncology Department, AUSL Romagna, 48100 Ravenna, Italy; anna.cariello@auslromagna.it; 34Radiation Oncology Department, IRCCS Regina Elena National Cancer Institute, 00144 Rome, Italy; giuseppe.sanguineti@ifo.it; 35Breast Surgery Department, IRCCS Regina Elena National Cancer Institute, 00144 Rome, Italy; claudio.botti@ifo.it (C.B.); fabio.pelle@ifo.it (F.P.); sonia.cappelli@ifo.it (S.C.); flavia.cavicchi@ifo.it (F.C.); ilaria.puccica@ifo.it (I.P.); amedeo.villanucci@ifo.it (A.V.); 36Biostatistics Unit, IRCCS Regina Elena National Cancer Institute, 00144 Rome, Italy; isabella.sperduti@ifo.it; 37Scientific Direction, IRCCS Regina Elena National Cancer Institute, 00144 Rome, Italy; gennaro.ciliberto@ifo.it

**Keywords:** breast cancer, HER2-positive subtype, adjuvant T-DM1, adverse events, real-world evidence

## Abstract

In patients with HER2-positive breast cancer, the goal of neoadjuvant therapy is to achieve a pathological complete response (pCR). However, half of patients have residual invasive disease at definitive surgery after neoadjuvant treatment. For these patients, T-DM1, a drug that combines trastuzumab with a chemotherapy agent, is employed in the adjuvant setting to reduce the risk of recurrence. This study primarily aimed to evaluate the tolerability of T-DM1 in these patients in a real-world setting, with a secondary focus on its effectiveness. Among 410 patients treated across multiple Italian cancer centers, more than half experienced side effects, with a small proportion developing more severe adverse events. In terms of effectiveness, the median follow-up period and number of recurrence events were insufficient to allow for a formal analysis. Consequently, extended follow-up is necessary to fully understand the long-term impact of T-DM1 in routine clinical practice.

## 1. Introduction

Human epidermal growth factor receptor 2 (HER2)-positive breast cancer (HER2^+^BC) represents approximately 15–20% of all breast tumors and is associated with aggressive behavior and poor prognosis [1]. These tumors are characterized by HER2 protein overexpression, measured by immunohistochemistry (IHC), or *HER2* amplification, detected by in situ hybridization (ISH). Anti-HER2 targeted therapies have dramatically altered the natural history of HER2^+^BC, improving outcomes in both early-stage and metastatic disease [2,3]. Locoregional surgery, systemic treatment, including neoadjuvant and adjuvant chemotherapy with anti-HER2 agents, endocrine therapy, and radiotherapy are all commonly employed in treating early HER2^+^BC. Neoadjuvant treatment (NAT) is commonly administered in cases of locally advanced and operable BC, with the primary objective of achieving a pathological complete response (pCR). In the case of HER2^+^BC, a pCR is strongly associated with significantly improved long-term outcomes [4,5], making it a key therapeutic goal. The combination of chemotherapy and trastuzumab has demonstrated significant advantages in achieving a pCR in the early HER2^+^BC setting. This approach has been shown to reduce the risk of relapse, disease progression, and death when compared to chemotherapy alone, offering a more effective treatment strategy for this group of patients [6,7]. Numerous efforts have been undertaken to increase pCR rates and improve long-term outcomes in early HER2^+^BC treatment. A pivotal multicenter, open-label, phase II randomized trial, NeoSphere, was the first study to demonstrate a significant improvement in pCR rates among patients receiving a combination of pertuzumab, trastuzumab, and docetaxel compared to those receiving either pertuzumab or trastuzumab plus docetaxel, or dual HER2 blockade, without chemotherapy. These findings led to the approval of pertuzumab in the neoadjuvant setting, marking a major advancement in treatment strategies for HER2^+^BC [8]. A pooled analysis further confirmed the more favorable event-free survival in patients who achieved a pCR after neoadjuvant treatment with dual HER2 blockade compared to those treated with trastuzumab alone [9].

Despite the advancements accomplished with HER2-targeted agents in the neoadjuvant setting, approximately 40–60% of HER2^+^BC patients who undergo standard neoadjuvant therapy have residual disease at surgery. These patients face a significantly higher risk of both local and systemic relapse [8,10], underscoring the need for more effective strategies to improve long-term outcomes. Trastuzumab emtansine (T-DM1) is an antibody–drug conjugate consisting of trastuzumab linked via a non-reducible thioether bond to a cytotoxic agent, emtansine, a microtubule inhibitor [11]. T-DM1 has demonstrated significant activity in patients with pretreated advanced HER2^+^BC [12], offering a targeted and potent therapeutic option for this population. In the early BC setting, additional post-neoadjuvant treatment with T-DM1 in patients with HER2^+^BC who did not achieve a pCR at surgery has been shown to significantly improve both invasive disease-free survival (iDFS) and overall survival (OS). These findings were demonstrated in the KATHERINE trial [13,14], establishing T-DM1 as the current standard of care for this patient population. In the safety analysis of the KATHERINE trial, a higher incidence of grade 3 or greater AEs was observed in patients treated with adjuvant T-DM1 (25.7%) compared to those receiving adjuvant trastuzumab (15.4%). Additionally, 18.0% of patients in the T-DM1 group discontinued treatment due to adverse events, highlighting the increased toxicity profile associated with T-DM1 in this setting [13].

Currently, real-world data on the effectiveness and tolerability of adjuvant T-DM1 remain limited. On this basis, we conducted a multicenter, observational, retrospective study to primarily assess the tolerability of adjuvant T-DM1, with a secondary objective of evaluating its effectiveness in HER2^+^BC patients who had undergone NAT and had invasive residual disease at surgery.

## 2. Materials and Methods

### 2.1. Study Approval

The ATD study was a multicenter, observational, retrospective investigation focused on early HER2^+^BC patients who exhibited invasive residual disease at surgery following neoadjuvant chemotherapy, including anti-HER2 therapies, and subsequently received T-DM1 in an adjuvant setting. This study was granted approval by the Institutional Review Board (IRB) of the coordinating center, the IRCCS Regina Elena National Cancer Institute of Rome, Italy [RS N 1480/21], as well as by the IRBs of the other recruiting centers. All procedures were conducted in strict compliance with the principles outlined in the Declaration of Helsinki. A total of 24 cancer centers across Italy collaborated for this study. Patients provided written informed consent, ensuring ethical and transparent involvement in the research.

### 2.2. Patient Selection

All patients included in the study were women diagnosed with HER2^+^BC. Specifically, the eligibility criteria required histologically confirmed, HER2^+^, non-metastatic, invasive primary BC, treated with neoadjuvant therapy (NAT) and demonstrating residual invasive disease in the breast and/or axillary lymph nodes following surgery. Pathological assessments were conducted by dedicated pathologists at each participating center. HER2 status was evaluated in both pretreatment biopsy samples and post-surgical specimens, adhering to the latest ASCO/CAP guidelines for HER2 testing [15]. HER2 positivity was defined as an IHC score of 3+ (DAKO Herceptest) or confirmed by in situ hybridization (FISH, CISH, or SISH) in cases with a HER2 IHC score of 2+. Hormone receptor expression (estrogen receptor [ER] and progesterone receptor [PgR]) and the Ki-67 proliferation index were also assessed by immunohistochemistry (IHC) in both pretreatment and post-surgical samples. Hormone receptor expression (ER and PgR) was considered significant when more than 1% of invasive tumor cells demonstrated positive immunostaining. The threshold for Ki-67 expression was set to 20% to define cases with a high proliferation rate (>20%). Both histological and biological characteristics were evaluated in the invasive component of the tumor. Tumor staging followed the TNM system (AJCC, 8th edition) [16], and histological grading used the Bloom–Richardson classification. Patients could have received trastuzumab in combination with taxane-based, anthracycline–taxane-based, or non-anthracycline–taxane-based chemotherapy as part of their neoadjuvant regimen. Pertuzumab was also allowed when available and indicated. Exclusion criteria included contraindications to T-DM1, the development of distant metastases during NAT, or a personal history of other malignant neoplasms.

### 2.3. Data Collection

Comprehensive restaging was performed for all patients following surgery, and T-DM1 treatment was administered intravenously every three weeks (21-day cycle) for up to 14 cycles, with an initial dose of 3.6 mg per kilogram of body weight. Treatment continued until either completion of the planned cycles, disease progression, unacceptable toxicity, physician recommendation, or patient refusal. The patients recruited had been receiving T-DM1 from May 2019 to the time of recruitement (September 2024). Adjuvant radiotherapy and/or endocrine therapy were provided when clinically indicated.

Clinical data were retrospectively extracted from medical records and included age at diagnosis, menopausal status, performance status at the time of diagnosis, the type and duration of prior NAT, treatment-related toxicities, types of surgery performed, definitive histological results, and the administration of radiotherapy or hormone therapy. Tumor characteristics and molecular profile were assessed on the original pathological report, when possible. The tolerability of T-DM1 was assessed by documenting adverse events (AEs) according to the Common Terminology Criteria for Adverse Events (CTCAE, version 5). Additionally, detailed follow-up data were collected to monitor time-to-event outcomes such as relapse-free survival (RFS) and OS. Median follow-up was calculated from the pathological diagnosis of the residual tumor after surgery to the date of recurrence, death, or the last follow-up. All data were encoded and entered into a dedicated database overseen by the coordinating center, with data entry performed by the recruiting centers.

### 2.4. Study Endpoints

The primary objective of the ATD study was to evaluate the tolerability of adjuvant T-DM1 treatment, measured in terms of AEs according to the CTCAE, version 5, in a real-world population. The secondary objective focused on assessing the effectiveness of adjuvant T-DM1 in HER2^+^BC patients who had undergone NAT and exhibited invasive residual disease at surgery in terms of RFS and OS measured from the time of pathological diagnosis of invasive residual disease.

### 2.5. Statistical Methods

The sample size calculation was based on the primary objective of assessing the toxicity profile and tolerability of adjuvant T-DM1, focusing on the proportion of patients experiencing grade 3 or greater adverse events. An expected grade ≥ 3 AE rate of 20% was assumed. To achieve 80% statistical power for detecting it, with a 3% margin of error and a 95% confidence level, a total of 291 patients were required (alpha 0.05). This sample size ensured sufficient power to reliably estimate the safety outcomes in the real-world population.

All variables included in the data collection forms were analyzed, with descriptive statistics provided for each: mean, median, standard deviation, range, minimum and maximum values for continuous variables, and absolute and relative frequencies for categorical variables. Associations between categorical variables were evaluated using the Chi-square test or Fisher’s exact test, as appropriate. Relapse-free events were defined as either disease recurrence or death from any cause. Overall survival events were defined as death from any cause. All statistical analyses were performed independently by two authors using SPSS statistical software (v21.0) and R programming (v4.4.1).

## 3. Results

### 3.1. Patient and Tumor Characteristics

From May 2019 to January 2024, 410 patients meeting the inclusion criteria were included in the present study. All patients received at least one cycle of adjuvant T-DM1, and at the time of this analysis, 18 (4.4%) were still undergoing treatment. The patient and tumor characteristics prior to NAT are summarized in Table 1, while details regarding neoadjuvant regimens, types of definitive surgery, and residual disease characteristics are presented in Table 2.

The median age at diagnosis was 51 years (range, 21–83), reflecting a broad age distribution, with a nearly equal split between premenopausal (50.5%) and postmenopausal (49.5%) patients. Invasive ductal carcinoma was the predominant histotype, observed in 84.4% of cases, consistent with the known prevalence of this subtype in BC. Invasive lobular carcinoma was present in 5.1% of cases, while 10.5% were classified as mixed or other histotypes. Tumors were primarily classified as cT1 (15.4%) or cT2 (62.2%), collectively comprising 77.6% of cases, while more advanced cT3 (10.2%) and cT4 (4.9%) tumors were less common. Clinical staging identified axillary node involvement in 52.3% of patients (cN1–cN3), a hallmark of more aggressive disease at presentation. The histological grading further reflected the aggressive nature of the disease, with over half of the tumors (55.6%) classified as G3, while 32.9% were G2, and only 0.5% were G1. HER2 positivity was universally confirmed in baseline biopsies, with the majority of cases (63.4%) showing an IHC score of 3+. Additionally, 30.7% of cases were classified as HER2 2+ by IHC with ISH amplification. Interestingly, a small subset of cases (0.7%) had HER2 amplification confirmed by ISH despite an IHC score of 1+ or 0, highlighting the importance of second-level molecular testing for accurate HER2 classification. Regarding hormone receptor expression, 77.8% of tumors were ER-positive, and 65.4% expressed PgR, with 64.4% of cases being triple-positive (HER2+, ER+, PgR+). Conversely, 21.0% of tumors were classified as HER2-enriched-like (ER- and PgR-negative), and 0.2% had an unreported hormone receptor status. Ki-67, a marker of proliferation, was high (>20%) in 74.6% of cases, further reinforcing the aggressive biological behavior of these tumors. These data emphasize the high prevalence of biologically aggressive, high-grade tumors in this cohort, consistent with the expected clinical profile of HER2+ breast cancer.

### 3.2. Neoadjuvant Treatment and Surgical Outcomes

All patients received prior systemic NAT. The majority, 345 (84.1%), were treated with an anthracycline–taxane-based regimen plus trastuzumab, while 20 (4.9%) received taxane plus trastuzumab. One patient (0.2%) underwent an anthracycline- and taxane-free regimen combined with trastuzumab, and thirty-five (8.5%) received trastuzumab/pertuzumab plus chemotherapy. The specific neoadjuvant regimen was not detailed for nine patients (2.2%). NAT was completed in 356 patients (86.8%) according to the preplanned number of cycles based on the selected regimen. Of the 54 patients (13.2%) who did not complete the treatment, 27 discontinued due to toxicity, 4 withdrew, 1 experienced local progression, and for the remaining 22 patients, the reason was not reported.

Following NAT, 177 patients (43.2%) underwent breast-conserving surgery, either a lumpectomy or quadrantectomy, while mastectomy was performed in 233 cases (56.8%). For regional lymph node management, a sentinel node biopsy was conducted in 192 patients (46.8%), and axillary dissection was performed in 169 patients (41.2%). The type of nodal surgery was not specified for 49 patients (12.0%). At the time of definitive surgery, all patients had residual disease in either the breast or regional lymph nodes, with 172 patients (42.0%) exhibiting residual node-positive disease. Regarding pT classification, 34 patients (8.3%) had no residual invasive disease in the breast (pT0/pTis), while the majority, 311 (75.9%), had ypT1 residual disease. Additionally, 55 patients (13.4%) had ypT2, 7 (1.7%) had ypT3, and 2 (0.5%) had ypT4 residual disease, while pT status was unspecified for 1 patient (0.2%). With respect to the baseline evaluation at biopsy, residual disease at surgery remained HER2^+^ in 319 cases (77.8%), changed into HER2^−^ in 18 (4.4%) cases, and was not reported for 73 (17.8%) cases. Of the 319 HER2^+^ residual tumor samples, 213 were found to be HER2 3+ by IHC, 92 were found to be HER2 2+ by IHC and ISH-amplified, 9 were diagnosed as HER2 1+ by IHC and ISH-amplified, and 5 were diagnosed as HER2 0 by IHC and ISH-amplified. In 18 cases (4.4%), the residual disease was discordant with the baseline HER2 status, becoming HER2-negative. Among this subset, the HER2 IHC score was 2+ in 9 cases, 1+ in 5 cases, and 0 in 4 cases, with all cases showing non-amplified ISH results. HER2 status was not fully characterized in the remaining 73 cases (17.8%), as ISH results were not reported. Within this group, the HER2 IHC scores were 2+ in 8 cases, 1+ in 15 cases, 0 in 20 cases, and unspecified in 30 cases. Among the 129 patients with HER2 ISH amplification at biopsy (and IHC scores of 2+, 1+, or 0), 66 retained HER2 amplification in the residual tumor. Within this group, the IHC score at surgery was 2+ in 58 cases, 1+ in 5 cases, 0 in 2 cases, and not reported in 1 case. Conversely, 12 cases lost HER2 ISH amplification in the residual disease, with corresponding IHC scores of 2+ in 6 cases, 1+ in 4 cases, and 0 in 2 cases. For the remaining 51 patients with HER2 ISH amplification at baseline, ISH was not performed on the surgical specimen. Among these patients, the IHC score was 3+ in 25 cases, 2+ in 4 cases, 1+ in 5 cases, 0 in 11 cases, and not reported in 6 cases. Regarding HR status, residual tumors were ER-positive in 297 patients (72.4%) and PgR-positive in 210 (51.2%). Both ER and PgR were positive in 204 patients (49.8%), while both receptors were negative in 83 cases (20.2%). HR status was not reported for at least one among ER and PgR in 24 cases (5.9%). For 385 patients (94.0%), ER and PgR status was available at both baseline and at residual disease evaluation. Among these, 352 cases (85.9% of the total study population) showed a concordant ER status, while 33 (8.1%) were discordant, with 18 cases changing from ER-positive to ER-negative and 15 cases changing from ER-negative to ER-positive. Similarly, for PgR status, 279 patients (68.0%) had concordant results at baseline and surgery. However, 74 cases (18.0%) shifted from PgR-positive to PgR-negative, and 32 cases (7.8%) changed from PgR-negative to PgR-positive.

### 3.3. Adjuvant Treatment with T-DM1

Following surgery, all patients received adjuvant treatments. Adjuvant radiation therapy was administered to 303 patients (73.9%), with 282 receiving it prior to T-DM1 and 21 undergoing radiotherapy concurrently with T-DM1. Endocrine therapy was given to 291 patients (71.0%), delivered either concomitantly with radiotherapy and/or during T-DM1 treatment. Data on adjuvant treatments are reported in Table 3.

The median time from surgery to the initiation of T-DM1 was 2 months (range, 1–7 months), with 205 patients (50.0%) receiving their first cycle within this 2-month window. The median number of T-DM1 cycles administered was 14 (range, 1–17). Specifically, 289 patients (70.5%) received the full 14 cycles of T-DM1, while 102 patients (24.9%) received between 1 and 13 cycles. Two patients (0.5%) received, respectively, 15 and 17 cycles, and for seventeen patients (4.1%), the number of cycles was not reported. At the time of this analysis, treatment was still ongoing for 18 patients (4.4%). For a total of 41 patients (10.0%), T-DM1 was discontinued, with the reason being explicitly reported as toxicity, 5 patients (1.2%) withdrew voluntarily, and 8 patients (2.0%) experienced a relapse during T-DM1 treatment. An additional 45 (10.9%) patients interrupted T-DM1 for unspecified reasons, and for 2 (0.5%) patients, information on T-DM1 completion (and number of cycles) was not reported.

### 3.4. Tolerability and Adverse Events

Overall, 228 patients (55.6%) experienced at least one adverse event (AE) associated with T-DM1. Grade 3 or 4 treatment-related AEs were observed in 20 patients (4.9%), including thrombocytopenia (n = 6), hepatotoxicity (n = 3), neutropenia (n = 2), anemia (n = 2), cardiotoxicity (n = 1), gastrointestinal toxicity (n = 1), neurotoxicity (n = 1), allergic reaction (n = 1), ocular toxicity (n = 1), pneumonitis (n = 1), and other toxicities (n = 1). A single grade 5 AE, a case of T-DM1-related pneumonitis, was reported. The comprehensive data on the safety of T-DM1 that we collected are given in Table 4.

The most frequent AEs, defined as those with an incidence of 5.0% or greater, included hepatotoxicity (18.5%), thrombocytopenia (17.6%), gastrointestinal toxicity (13.2%), neurotoxicity (6.3%), and neutropenia (5.6%). Cardiotoxicity related to T-DM1 was reported in 2.4% of patients. We report these main toxicities with the respective distribution of grades in Figure 1.

Other toxicities, such as fatigue, mucositis, nausea, anemia, ocular toxicity, pneumonitis, allergic reactions, and fever, each occurred in fewer than 2.0% of cases. Unspecified toxicities, registered as “other toxicities,” were reported in 31 patients (7.6%), with only 1 case being grade 3 and none reaching grade 4 or 5 severity.

We conducted a comparative analysis between the 291 patients who completed at least 14 cycles of T-DM1 and those who discontinued treatment after receiving between 1 and 13 cycles. To ensure a valid comparison, we excluded patients with ongoing T-DM1 treatment (n = 18), those who experienced a relapse during T-DM1 therapy (n = 8), and cases where data on T-DM1 completion and cycle number were both missing (n = 2). This resulted in a cohort of 91 patients who discontinued T-DM1 due to either toxicity (n = 41), voluntary withdrawal (n = 5), or unspecified reasons (n = 45). The incidence of any-grade adverse events (AEs) was 55.3% (161/291) among those who completed T-DM1, compared to 61.5% (56/91) in patients who interrupted treatment (Chi-square test, *p* = 0.36). For grade 3 or higher AEs, the incidence was significantly lower in the patients who completed treatment, at 3.1% (9/291), versus 12.1% (11/91) in those who interrupted T-DM1 (Chi-square test, *p* = 0.002). When restricting the comparison to the 45 patients who discontinued T-DM1 for unspecified reasons, the incidence of any-grade AEs was 33.3% (15/45) versus 53.3% (161/291) in the patients who completed treatment. This discrepancy suggests an underreporting of toxicities in the subset of patients for whom the reason for discontinuation was not specified, as well as in the reporting of the cause for interruption itself. In contrast, among the 41 patients for whom T-DM1 discontinuation was explicitly linked to toxicity, 95.1% (31/41) experienced at least one AE of any grade, with 11 cases involving grade 3 or higher toxicities. The most common AEs in this group were thrombocytopenia (34.1% of any grade, with 4.9% grade 3), hepatotoxicity (34.1% of any grade, with 4.9% grade 3, including one case with concurrent grade 3 thrombocytopenia), gastrointestinal toxicity (26.8% of any grade, with no grade 3 or higher events), and neurotoxicity (9.9% of any grade, with 2.4% grade 3).

### 3.5. Preliminary Outcomes of Relapse and Survival

At the time of this analysis, the median follow-up from the pathological diagnosis of the residual tumor post-surgery was 25 months (range, 1–55 months). During this follow-up period, 31 relapse events (7.6% of patients) were recorded. Of these, 4 relapses occurred in the ipsilateral breast, 1 in the contralateral breast, and 22 at distant sites; in 4 cases, the secondary site was not specified. A total of 22 deaths (5.4%) from any cause were also reported. The follow-up period, however, was not sufficient, and the number of events was too limited to allow for formal survival analyses of either time-to-relapse or time-to-death outcomes. Nonetheless, we performed non-parametric comparisons using Fisher’s exact test to evaluate the distribution of events across relevant patient groups. Patients who were still undergoing T-DM1 treatment (n = 18) or those who experienced relapse during T-DM1 treatment were excluded from the analysis, as these conditions represented competing factors that could confound the potential impact of incomplete T-DM1 administration on relapse and mortality rates. Additionally, two patients for whom data on T-DM1 completion and the number of administered cycles were both missing were also excluded.

After these exclusions, the incidence of relapse was 4.4% (4/91) in patients who received fewer than 14 cycles of T-DM1, compared to 6.2% (18/291) in those who completed at least 14 cycles. The odds ratio for relapse between patients who discontinued T-DM1 and those who completed treatment was 0.70 (*p* = 0.616), suggesting no significant difference. Similarly, the incidence of death from any cause was 3.3% (3/91) in the group that discontinued treatment, compared to 4.5% (13/291) in those who completed the full course of T-DM1, with an odds ratio of 0.73 (*p* = 0.771), again indicating no statistically significant difference.

## 4. Discussion

In this multicenter, observational, retrospective study, we evaluated the tolerability of adjuvant T-DM1 in a real-world population of 410 patients with HER2^+^BC who had residual invasive disease after NAT. Our primary objective was to assess the safety profile of T-DM1 in this setting, while the secondary objective was to provide data on its effectiveness in terms of RFS and OS.

Our findings demonstrate that adjuvant T-DM1 is generally well tolerated in routine clinical practice. We observed that 55.6% of patients experienced at least one AE associated with T-DM1. Grade 3 or 4 treatment-related AEs occurred in 4.9% of patients, and only one grade 5 AE (pneumonitis) was reported. Lung toxicity, including pneumonitis, interstitial lung disease, and acute respiratory distress syndrome, is a relatively rare event associated with T-DM1 treatment. An integrated safety analysis of phase III trials reported an incidence of 1.1% of such toxicity, with death occurring in approximately 0.1% of cases [17]. Our findings are consistent with these reports, with a total of two cases of lung toxicity observed, including the grade 5 pneumonitis case mentioned and an additional case of grade 3 pneumonitis. Interstitial lung disease is of particular concern for the class of antibody–drug conjugates, which T-DM1 belongs to. However, this incidence is significantly higher with the use of other compounds of the same category such as trastruzumab deruxtecan (around 15%) and trastuzumab duocarmazine (around 7%) [18]. The most frequent AEs (incidence ≥ 5%) in this study were hepatotoxicity (18.5%), thrombocytopenia (17.6%), gastrointestinal toxicity (13.2%), neurotoxicity (6.3%), and neutropenia (5.6%). This profile also aligns with findings from the previously mentioned integrated safety analysis, which included 834 patients across six randomized trials conducted in various treatment settings [17]. Importantly, 70.5% of patients received the full 14 cycles of T-DM1, and 10% discontinued treatment explicitly due to toxicity. When comparing our results with the pivotal KATHERINE trial [13], several similarities and differences emerge. In KATHERINE, 71.4% of patients completed all 14 cycles of T-DM1, which is comparable to the 70.5% completion rate in our study. However, the incidence of grade 3 or higher AEs was significantly higher in KATHERINE, occurring in 25.7% of patients receiving T-DM1, compared to 4.9% in our cohort. Additionally, adverse events leading to the discontinuation of T-DM1 occurred in 18.0% of patients in KATHERINE, whereas only 10% of patients in our study discontinued T-DM1 due to toxicity. The most common grade ≥3 AEs in KATHERINE were thrombocytopenia (5.7%) and hypertension (2.0%), while in our study, thrombocytopenia and hepatotoxicity were the most frequent grade ≥3 AEs but occurred at lower rates (each in 1.5% of patients). A specific area of interest relates to the incidence and reporting of neuropathy, a potential long-term toxicity. In our cohort, neuropathy of any grade was observed in 6.3% of patients, with grade 2 or higher neuropathy reported in 2.9%. These findings align with rates reported in the real-world study KARMA [19] (7.9%) and the KATHERINE trial, where neuropathy of any grade was less than 10%. However, the ATEMPT trial [20], conducted in a different setting (adjuvant after upfront surgery), reported a higher incidence of grade 2 or greater neuropathy (11%) compared to our cohort. Furthermore, a meta-analysis indicated a 1.6% rate of grade 3 or greater neuropathy with T-DM1 across various settings, which is significantly higher than the 0.2% incidence of grade 3 neuropathy observed in our study [21]. This discrepancy may reflect the inherent underreporting associated with retrospective designs. These findings highlight the importance of systematic and comprehensive toxicity documentation, particularly for persistent and impactful side effects like neuropathy, which can have long-term consequences on patient quality of life. The KARMA study [19], conducted in Spain, reported that only 70.2% of patients were still under adjuvant treatment at the time of analysis, with a median of six cycles of T-DM1 administered. In contrast, our study achieved a median of 14 cycles administered. Treatment-related grade 3 AEs in KARMA were reported in 5.3% of patients, similar to the 4.9% observed in our study and notably lower than the 25.7% in KATHERINE. No grade 4 or 5 AEs related to T-DM1 were reported in KARMA, whereas we reported one grade 5 AE (pneumonitis).

The lower incidence of severe AEs in our study and KARMA compared to KATHERINE may be attributed to differences between randomized controlled trials (RCTs) and real-world settings [22,23]. RCTs like KATHERINE often involve more rigorous monitoring and comprehensive reporting of AEs, potentially leading to higher reported toxicity rates. In real-world practice, clinicians may exercise more flexibility in managing treatment-related toxicities, including dose modifications or delays, which may reduce the incidence of severe AEs. Additionally, patient populations in real-world studies may differ from those in RCTs due to broader inclusion criteria and variations in clinical characteristics. Thus, the main elements that change from RCTs to real-world studies include patient selection, treatment administration, and monitoring practices. RCTs often have strict eligibility criteria, resulting in a highly selective patient population with potentially better overall health and fewer comorbidities. In contrast, real-world studies encompass a broader spectrum of patients, including those with comorbidities or varying performance statuses, which can impact both the safety and effectiveness outcomes observed. Additionally, real-world clinicians may tailor treatment regimens based on individual patient needs, which can influence adherence and toxicity profiles.

Notably, pertuzumab pretreatment rates also varied among our data and the other studies. In KATHERINE, 18% of patients received pertuzumab in combination with trastuzumab in the neoadjuvant phase. In our study, only 8.5% of patients received pertuzumab pretreatment, reflecting the national guidelines and availability during the study period. In contrast, the KARMA study reported a higher pertuzumab pretreatment rate of 86.8%. The lower pertuzumab use in our cohort may have influenced the tolerability and management of subsequent T-DM1 therapy, although further investigation is needed to elucidate this relationship.

Our study has several strengths. It represents one of the largest real-world cohorts examining the safety of adjuvant T-DM1 in patients with HER2^+^BC and residual disease after NAT. Its multicenter nature and the inclusion of multiple cancer centers across Italy enhance the generalizability of our findings. Additionally, the high rate of completion of the planned 14 cycles of T-DM1 underscores the feasibility of administering this therapy in routine practice. However, there are limitations to consider. The retrospective design may introduce biases, including underreporting of AEs and incomplete data capture. Our data model for toxicity reporting prioritized key adverse event categories to streamline data collection and enhance compliance across centers, but the inclusion of an open-ended ‘Other toxicities’ field, while useful for capturing rare events, may have contributed to underreporting due to the frequent lack of specification and the limitation of recording only one such event per patient. Furthermore, the multicenter nature of the study, while enhancing its generalizability, may have introduced variability into the data collection and reporting practices used across the participating centers. These factors could have impacted the consistency and completeness of the dataset, representing potential sources of error. Our study was adequately powered to evaluate the primary endpoint of tolerability, with the sample size exceeding the requirements for detecting AEs at a 20% rate. However, the relatively short median follow-up of 25 months and the limited number of relapse events preclude definitive conclusions regarding the effectiveness of T-DM1 in this setting. Our secondary and exploratory analyses are preliminary, and longer follow-up is necessary to assess long-term outcomes such as RFS and OS. Finally, the absence of routinely collected data on ethnicity in Italian clinical practice limits the ability to explore potential differences in outcomes across diverse racial groups, although the majority of patients are presumed to be Caucasian based on national demographics.

## 5. Conclusions

In conclusion, our study suggests that adjuvant T-DM1 is well tolerated in a real-world population of patients with HER2^+^BC and residual invasive disease after neoadjuvant treatment. The safety profile observed is consistent with that reported in previous studies, including the prospective randomized KATHERINE trial, which compared T-DM1 to trastuzumab, and the KARMA study, which reported data on the efficacy and safety of T-DM1 in a real-world setting. These findings support the use of T-DM1 as a standard adjuvant treatment in this patient population, although continued monitoring and further research are necessary to fully understand its long-term impact. An updated analysis with extended follow-up will be important to validate our preliminary findings on the effectiveness of adjuvant T-DM1. As more events accrue over time, we will be able to perform robust survival analyses and potentially identify factors influencing outcomes. Further research could also explore the impact of variables such as hormone receptor status and HER2 expression changes and specific toxicity profiles on the tolerability and effectiveness of T-DM1.

## Figures and Tables

**Figure 1 cancers-16-04104-f001:**
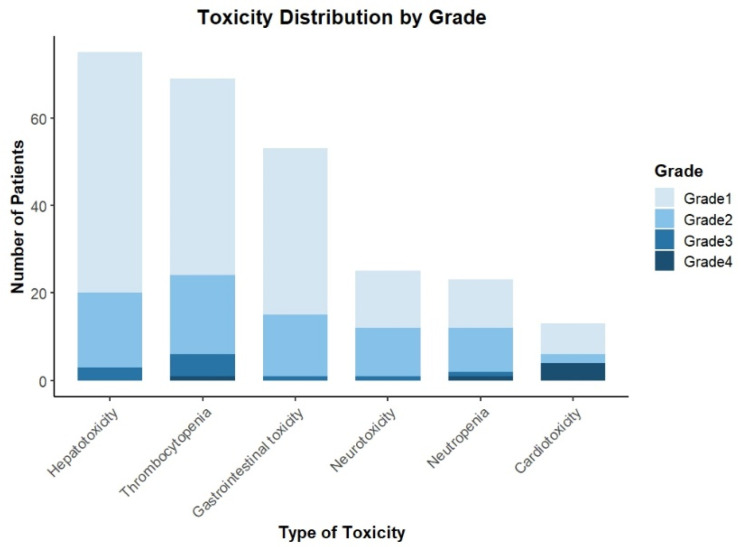
The distribution of the most common toxicities with respective grades.

**Table 1 cancers-16-04104-t001:** Patient and tumor characteristics prior to neoadjuvant treatment.

		Median	Range
Age at Diagnosis		51	21–83
		Absolute Count	Percentage
Menopause			
	Premenopause	207	50.5
	Postmenopause	203	49.5
Clinical T size			
	1	63	15.4
	2	255	62.2
	3	42	10.2
	4	20	4.9
	Unknown	30	7.3
Clinical N involvement			
	0	167	40.7
	1	145	35.4
	2	61	14.9
	3	8	2
	Unknown	29	7.1
Histotype			
	Ductal	346	84.4
	Lobular	21	5.1
	Other	29	7.1
	Ductal-Lobular	5	1.2
	Unknown	9	2.2
Grade			
	1	2	0.5
	2	135	32.9
	3	228	55.6
	Unknown	45	11
ER			
	Negative	90	22
	Positive	319	77.8
	Unknown	1	0.2
PgR			
	Negative	141	34.4
	Positive	268	65.4
	Unknown	1	0.2
HER2 IHC + score			
	3	260	63.4
	2	126	30.7
	1	2	0.5
	0	1	0.2
	Unknown	21	5.1
HER2 ISH			
	Amplification	163	39.8
	Not done	247	60.2
HER2 category			
	Positive	410	100
Ki-67			
	Low	75	18.3
	High	306	74.6
	Unknown	29	7.1

Abbreviations: clinical T size = the size of the primary tumor based on clinical findings (physical examination and imaging modalities, such as mammography, ultrasound, and MR imaging), where 1 corresponds to ≤20 mm, 2 corresponds to >20 mm but ≤50 mm, 3 corresponds to >50 mm, and 4 corresponds to a primary tumor of any dimensions with direct extension to the chest wall or to the skin; clinical N involvement = clinical extension of the tumor in the regional lymph nodes, where 1 indicates involvement of level I–II, 2 indicates matted lymph nodes at levels I–II or involvement of ipsilateral internal mammary nodes in the absence of axillary lymph node metastases, and 3 indicates concomitant involvement of levels I–II and ipsilateral internal mammary nodes or involvement of level III.

**Table 2 cancers-16-04104-t002:** Neoadjuvant regimens employed, the type of definitive surgery, and residual disease characteristics.

		Absolute Count	Percentage
Neoadjuvant Treatment			
	Chemotherapy plus Pertuzumab–Trastuzumab	35	8.5
	Anthracycline–Taxane plus Trastuzumab	345	84.1
	Taxane plus Trastuzumab	20	4.9
	Other Chemotherapy plus Trastuzumab	1	0.2
	Unknown	9	2.2
Neoadjuvant Treatment Completion			
	Completed	356	86.8
	Interrupted for Toxicity	27	6.6
	Interrupted for Progression	1	0.2
	Withdrew	4	1
	Unknown	22	5.4
Type of Breast Surgery			
	Mastectomy	233	56.8
	Conservative	177	43.2
Type of Nodal Surgery			
	Sentinel Lymph Node	192	46.8
	Axillary Lymph Node Dissection	169	41.2
	None	47	11.5
	Unknown	2	0.5
Pathological T Size			
	0	34	8.3
	1	311	75.9
	2	55	13.4
	3	7	1.7
	4	2	0.5
	Unknown	1	0.2
Pathological N Involvement			
	0	238	58
	1	118	28.8
	2	36	8.8
	3	18	4.4
Grade			
	1	21	5.1
	2	147	35.9
	3	172	42
	Unknown	70	17.1
ER			
	Negative	89	21.7
	Positive	297	72.4
	Unknown	24	5.9
PgR			
	Negative	176	42.9
	Positive	210	51.2
	Unknown	24	5.9
HER2 IHC + score			
	3	213	52
	2	109	26.6
	1	29	7.1
	0	26	6.3
	Unknown	33	8
HER2 ISH			
	Amplification	106	25.9
	No Amplification	18	4.4
	Not Performed/Unknown	286	69.8
HER2 Category			
	Positive	319	77.8
	Negative	18	4.4
	Unknown	73	17.8
Ki-67			
	Low	236	57.6
	High	133	32.4
	Unknown	41	10

**Table 3 cancers-16-04104-t003:** Description of adjuvant treatments.

		Absolute Count	Percentage
Adjuvant Hormone Therapy			
	Yes	291	71
	No	81	19,8
	Unknown	38	9,3
Adjuvant Radiotherapy			
	Yes	303	73,9
	No	101	24,6
	Unknown	6	1,5
Performance Status Pre-TDM1			
	0	373	91
	1	36	8,8
	Unknown	1	0,2
Number of TDM1 Cycles			
	14	289	70,5
	1 to 13	102	24,9
	15 to 17	2	0,5
	Unknown	17	4,1
TDM1 Completion			
	Completed	291	71
	Interrupted for Toxicity	41	10
	Interrupted for Relapse	8	2
	Withdrew	5	1,2
	Ongoing	18	4,4
	Interrupted for Unspecified Reason	45	10,9
	Unknown	2	0,5
		Median	Range
TDM1 Number of Cycles	Median	14	1–17

**Table 4 cancers-16-04104-t004:** Safety of T-DM1.

		Absolute Count	Percentage
Toxicity of any grade			
	Yes	228	55.6
	No	182	44.4
Cardiotoxicity			
	Yes	10	2.4
	No	400	97.6
Cardiotoxicity grade			
	1	7	1.7
	2	2	0.5
	4	1	0.2
	None/Unknown	400	97.6
Neutropenia			
	Yes	23	5.6
	No	387	94.4
Neutropenia grade			
	1	11	2.7
	2	10	2.4
	3	1	0.2
	4	1	0.2
	None/Unknown	387	94.4
Thrombocytopenia			
	Yes	72	17.6
	No	338	82.4
Thrombocytopenia grade			
	1	45	11
	2	18	4.4
	3	5	1.2
	4	1	0.2
	None/Unknown	341	83.2
Gastrointestinal toxicity			
	Yes	54	13.2
	No	356	86.8
Gastrointestinal toxicity grade			
	1	38	9.3
	2	14	3.4
	3	1	0.2
	None/Unknown	357	87.1
Neurotoxity			
	Yes	26	6.3
	No	384	93.7
Neurotoxicity grade			
	1	13	3.2
	2	11	2.7
	3	1	0.2
	None/Unknown	385	93.9
Hepatotoxicity			
	Yes	76	18.5
	No	334	81.5
Hepatotoxicity grade			
	1	55	13.4
	2	17	4.1
	3	3	0.7
	None/Unknown	335	81.7
Other toxicities			
	Yes	55	7.3
	No	355	86.6
Other toxicity type			
	Fatigue	7	1.7
	Mucositis	5	1.2
	Nausea	3	0.7
	Anemia	2	0.5
	Conjunctivitis	2	0.5
	Pneumonitis	2	0.5
	Allergy	1	0.2
	Fatigue	1	0.2
	Fever	1	0.2
	Unspecified	31	7.6
	No	355	86.6
Other toxicity grade			
	1	33	8
	2	10	2.4
	3	7	1.7
	5	1 (Pneumonitis)	0.2
	None/Uknown	359	87.6

## Data Availability

The data used to generate the results of the present study are available upon reasonable request, pending approval from all participating centers.

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
