# Peer review of "Tolerability and Preliminary Outcomes of Adjuvant T-DM1 in HER2-Positive Breast Cancer After Neoadjuvant Therapy: The ATD Study"

_cancers, 2024, doi:10.3390/cancers16234104_

Round 1
Reviewer 1 Report
Comments and Suggestions for Authors
Breast cancer poses a significant challenge, and further investigation into potential therapeutic methods is essential. The manuscript's topic is pertinent to cancer research and is generally well-written; however, several improvements are recommended.
The patient characteristics section should include the race of the subjects.
The presentation of the results and the manuscript's topic should be more detailed. The manuscript currently cites only 19 references, which may be inadequate for the scope of the research.
Minor
In the author contributions is used more format styles.
Author Response
Breast cancer poses a significant challenge, and further investigation into potential therapeutic methods is essential. The manuscript's topic is pertinent to cancer research and is generally well-written; however, several improvements are recommended.
- Comment 1: The patient characteristics section should include the race of the subjects.
- Response 1: This is an insightful observation by the reviewer. However, in Italy, information on ethnicity or race is not routinely collected in clinical practice or in medical records. As a result, this information is unavailable for our study cohort. Based on general population demographics in Italy, it is reasonable to estimate that the majority of patients in our study were Caucasian, with less than 10% representing other ethnic backgrounds, including African, Hispanic, or Asian descent. While this limitation is acknowledged, it is unlikely to significantly impact the generalizability of our findings within the context of Italian clinical practice. This aspect has been added to the limitations section of the revised manuscript.
- Comment 2: The presentation of the results and the manuscript's topic should be more detailed. The manuscript currently cites only 19 references, which may be inadequate for the scope of the research.
- Response 2: We acknowledge the reviewer's observation and agree that additional detail and references were warranted to better support the scope of our research. The descriptive nature of our results may have contributed to the perception of brevity, as inferential analyses were limited due to the low incidence of relapse and overall survival events in our cohort, which precluded formal statistical evaluations. We have now slightly extended the Results in section 3.1 (Patient and tumor characteristics) and section 3.2 in relation to the molecular changes in HER2 expression in residual disease compared to the baseline condition and in the section presenting basal patient and disease characteristics. Additionally, we have expanded the discussion section by incorporating further contextual analysis of our findings, including comparisons with relevant literature. These revisions have necessitated the inclusion of 4 additional references, enhancing the manuscript’s depth and alignment with the scope of the study. We appreciate the reviewer's valuable feedback, which has allowed us to improve the comprehensiveness of our manuscript.
- Comment 3: Minor: In the author contributions is used more format styles.
- Response 3: The author contributions section has been revised to ensure consistent formatting. Thank you for bringing this to our attention.
Reviewer 2 Report
Comments and Suggestions for Authors
Manuscript ID: cancers-3299394, Tolerability and outcomes of Adjuvant T-DM1 in HER2-positive breast cancer after Neoadjuvant therapy: The ATD Study
The article present the results of a multicenter, retrospective study across 24 oncology centers, including 410 patients with HER2+BC treated with adjuvant T-DM1. The manuscript follows the general format of such studies and is well organized and easy to follow.
The editing needs to be corrected as for the journal’s template. See “Cancers 2021, 13, x”, the tables titles, the references style, and others.
The section 2.2. could be expended to be easier to understand. Detail a little more on how the data presented in table 1 were collected or analyzed.
In table 2, the authors could simply write “unknown”, and not only uk. There is enough space in each cell of the table. The same for table 3.
The section 244 to 255 presents again the data from table 1. The authors should better comment on them or present something new, not but repeat it.
The discussion section should be more objective and critical. It should add more than a simple repletion of the results. What are the possible sources of error? Was the number of patients statistically significant? Are there AEs know as important for similar drugs that are grave but with a low frequency, AEs that could have been missed because the occurrence may be below 0.1? What other types of errors could have been?
The manuscript could use some schemes to better present the succession of events and overall, to better convey the information.
Author Response
The article present the results of a multicenter, retrospective study across 24 oncology centers, including 410 patients with HER2+BC treated with adjuvant T-DM1. The manuscript follows the general format of such studies and is well organized and easy to follow.
- Commen1: The editing needs to be corrected as for the journal’s template. See “Cancers 2021, 13, x”, the tables titles, the references style, and others.
- Response 1: Thank you for pointing this out. We have carefully revised the manuscript to ensure compliance with the journal’s template. Additionally, we reviewed the entire manuscript to address any remaining formatting inconsistencies.
- Comment 2: The section 2.2. could be expended to be easier to understand. Detail a little more on how the data presented in table 1 were collected or analyzed.
- Response 2: We appreciate the reviewer’s observation and have revised the manuscript to improve clarity and address this comment. Specifically, we have edited the original section 2.2 and split it into two sections: 2 Patient Selection and 2.3 Data Collection. This restructuring allows for a clearer distinction between how patients were selected for the study and how data were collected and managed. In the new Data Collection section, we have added further details on the data collection process, particularly regarding Table 1. Specifically, we included the following sentence to clarify how tumor characteristics and molecular profiles were gathered: "Tumor characteristics and molecular profile were assessed on the original pathological report, when possible." Regarding the analysis of these data, we have already provided this information in the Statistical Methods section (previously 2.4, now 2.5). This section describes the approach used to analyze the data, including the application of descriptive statistics for continuous and categorical variables, as follows: "All variables included in the data collection forms were analyzed, with descriptive statistics provided for each: mean, median, standard deviation, range, minimum and maximum values for continuous variables, and absolute and relative frequencies for categorical variables." These changes ensure that the data collection and analysis processes are clearly described, addressing the reviewer’s concern. We hope this revision meets your expectations.
- Comment 3: In table 2, the authors could simply write “unknown”, and not only uk. There is enough space in each cell of the table. The same for table 3.
- Response 3: Thank you for the suggestion. We have made the requested modification and replaced all instances of "uk" with "unknown" in Tables 2, 3, and all other tables throughout the manuscript.
- Comment 4: The section 244 to 255 presents again the data from table 1. The authors should better comment on them or present something new, not but repeat it.
- Response 4: Thank you for your valuable suggestion. We have revised the mentioned paragraph in the results section 3.1 to better contextualize and comment on the data presented in Table 1. Specifically, we expanded the narrative to include additional insights into the clinical and molecular profiles of the cohort, such as the prevalence of high-grade tumors, the significance of hormone receptor expression patterns, and the implications of proliferation markers like Ki-67. These additions provide a more detailed data presentation, avoiding mere repetition and offering some interpretative insights.
- Comment 5: The discussion section should be more objective and critical. It should add more than a simple repletion of the results. What are the possible sources of error? Was the number of patients statistically significant? Are there AEs know as important for similar drugs that are grave but with a low frequency, AEs that could have been missed because the occurrence may be below 0.1? What other types of errors could have been?
- Response 5: Thank you for this constructive feedback. We agree that the discussion can benefit from additional critical analysis. In response, we have revised the discussion section to address the following points:
Possible sources of error: We have expanded the discussion to include additional sources of potential error. Specifically, we acknowledge that the retrospective design of the study could introduce bias, particularly in the collection and reporting of adverse events and clinical data. Additionally, the multicenter nature of the study, while enhancing generalizability, may have resulted in variability in data recording practices. These factors may impact the completeness and accuracy of the dataset. (lines 497-505 in the clean version of the revised Manuscript)
Severe AEs known for similar drugs: We have incorporated a discussion of an important potential grave AEs associated with antibody-drug conjugates, which is, lung toxicity, including interstitial lung disease and pneumonitis. It is a known class effect of ADCs. We referenced incidence rates for related drugs, such as trastuzumab deruxtecan and trastuzumab duocarmazine, highlighting that while our observed rates align with published data for T-DM1, and are lower than those (lines 422-431 in the clean version of the revised Manuscript)
AEs with frequencies below 0.1%: We addressed the potential for missing AEs with low frequencies. To simplify reporting and improve compliance across centers , our data model for toxicity reporting focused on key categories such as cardiotoxicity, neutropenia, thrombocytopenia, gastrointestinal toxicity, neurotoxicity, and hepatotoxicity. However, we also provided an open-ended “Other toxicities” field to capture additional events which could have lower frequencies, which could be specified in free text. This approach may have led to underreporting, particularly as unspecified toxicities were common (33 out of 55 cases in this category). Furthermore, the model allowed only one entry for “Other toxicities” per patient, which could further limit the capture of rare AEs. (lines 497-505 in the clean version of the revised Manuscript)
Statistical significance of the sample size: We confirmed that the study was adequately powered to assess the primary endpoint of tolerability, with a sample size of 410 patients exceeding the required number for detecting adverse events with 20% rate with sufficient precision. (lines 505-507 in the clean version of the revised Manucript).
- Comment 6: The manuscript could use some schemes to better present the succession of events and overall, to better convey the information.
- Response 6: Thank you for this suggestion. Given the retrospective nature of our study and the specific eligibility criteria applied, the included patients were already well-defined as belonging to the post-neoadjuvant category with residual disease, eligible to receive T-DM1. Therefore, we determined that a detailed schematic representation of patient flow was not essential to the manuscript. While attrition tables were not generated, we recognize that a small number of cases (28 out of 438) were excluded during data merging due to failure to meet eligibility criteria. However, since this was the only point of attrition, we reasoned that it would not justify a full schematic representation. We hope this explanation clarifies our rationale for not including a scheme, but we remain open to adding such a figure if you believe it would further improve the clarity of the manuscript. Please let us know if you have specific suggestions in this regard.
Reviewer 3 Report
Comments and Suggestions for Authors
The research demonstrates the multi-center retrospective study of 410 patients with HER2-positive breast cancer treated with adjuvant trastuzumab emtansine. Trastuzumab emtansine demonstrated a manageable safety profile.
1. Table 1 and Table 2 need to be revised to indicate the differences between NA and unknown and all abbreviations in annotations. Clinical T size and N involvement need to be explained in the annotation of Table 1.
2. The results and percentages of toxicities of the trastuzumab emtansine are better to be visualized in graphs, not only in the table.
3. The conclusions may be revised to briefly explain the meanings of NAT, KATHERINE, and KARMA to emphasize the safety of trastuzumab emtansine.
Author Response
The research demonstrates the multi-center retrospective study of 410 patients with HER2-positive breast cancer treated with adjuvant trastuzumab emtansine. Trastuzumab emtansine demonstrated a manageable safety profile.
- Comment 1: Table 1 and Table 2 need to be revised to indicate the differences between NA and unknown and all abbreviations in annotations. Clinical T size and N involvement need to be explained in the annotation of Table 1.
- Response 1: Thank you for this helpful comment. We have revised both Table 1 and Table 2 to improve clarity and ensure consistency. Specifically, we replaced "NA" with "Not done" to more accurately reflect its intended meaning, and all instances of "uk" were replaced with the full term "unknown." Additionally, we expanded the annotations for Table 1 to include detailed explanations for "Clinical T size" and "N involvement," providing greater clarity for readers.
- Comment 2: The results and percentages of toxicities of the trastuzumab emtansine are better to be visualized in graphs, not only in the table.
- Response 2: Thank you for this valuable suggestion. In response, we have included a new figure (Figure 1) to enhance the visualization of toxicity data. This figure presents the most common toxicities as bar charts, with the respective grade incidence (Grades 1–4) stacked within each bar.
- Comment 3: The conclusions may be revised to briefly explain the meanings of NAT, KATHERINE, and KARMA to emphasize the safety of trastuzumab emtansine.
- Response 3: Thank you for the suggestion. We have revised the conclusions to briefly explain the meanings of NAT (neoadjuvant therapy), KATHERINE (a pivotal trial evaluating trastuzumab emtansine in HER2+ breast cancer), and KARMA (a real-world study on adjuvant T-DM1) to better emphasize the safety profile of trastuzumab emtansine. We believe these additions enhance clarity and reinforce the key findings of the study. Please let us know if further adjustments are required.